# Interactions between the Exocrine and the Endocrine Pancreas

**DOI:** 10.3390/jcm13041179

**Published:** 2024-02-19

**Authors:** Roberto Valente, Alessandro Coppola, Chiara Maria Scandavini, Asif Halimi, Annelie Magnusson, Augusto Lauro, Ira Sotirova, Urban Arnelo, Oskar Franklin

**Affiliations:** 1Department of Surgical and Perioperative Sciences, Surgery, Umeå University, 90185 Umeå, Sweden; roberto.valente@regionvasterbotten.se (R.V.); chiara.scanda@hotmail.it (C.M.S.); asif.halimi@regionvasterbotten.se (A.H.); annelie.magnusson@regionvasterbotten.se (A.M.); ira.sotirova@gmail.com (I.S.); urban.arnelo@regionvasterbotten.se (U.A.); oskar.franklin@cuanschutz.edu (O.F.); 2Department of Surgery, Division of Surgical Oncology, University of Colorado School of Medicine, Aurora, CO 80045, USA; 3Department of Surgery, Sapienza University of Rome, 00161 Rome, Italy; augusto.lauro@uniroma1.it

**Keywords:** diabetes, pancreatic exocrine insufficiency, interactions, pancreatic endocrine insufficiency, chronic pancreatitis, pancreas physiology

## Abstract

The pancreas has two main functions: to produce and secrete digestive enzymes (exocrine function) and to produce hormones that regulate blood glucose and splanchnic secretion (endocrine function). The endocrine and exocrine portions of the pancreas are central regulators in digestion and metabolism, with continuous crosstalk between their deeply interconnected components, which plays a role in disease. Pancreatic neoplasms, inflammation, trauma, and surgery can lead to the development of type 3c diabetes when an insult simultaneously damages both acini and islets, leading to exocrine and endocrine dysfunction. In diabetes mellitus patients, pancreatic exocrine insufficiency is highly prevalent, yet little is known about the associations between diabetes mellitus and pancreatic exocrine function. This review aims to provide an overview of the physiology of the pancreas, summarize the pathophysiology and diagnostic work-up of pancreatic exocrine insufficiency, and explore the relationships between exocrine pancreatic insufficiency and diabetes mellitus.

## 1. Introduction

The pancreas is an organ located in the retroperitoneum, behind the stomach, in close proximity to the duodenum, the hepatic hilum, and the spleen. Since ancient times, it has sparked curiosity both for its anatomy and its functions. In fact, understanding the anatomy of the pancreas began to improve only after the studies conducted by Johann Wirsung in Italy in 1642. This was followed by studies by Abraham Vater and Giovanni Santorini, providing a comprehensive picture of pancreatic anatomy by the mid-1700s. As for the function of the pancreas, we had to wait until the late 1800s. In 1850, Claude Bernard hypothesized that the pancreas was involved in digestion, and it was only in 1890 that Paul Langerhans hypothesized the role of the pancreas in diabetes [1].

The study of interactions between these two components of the pancreas, the exocrine and endocrine systems, still plays a significant role in pancreatic research. As evidence of this, in 2022, a conference was held at the National Institutes of Health (NIH) Natcher Conference Center in Bethesda, MD, hosted by the National Institute of Diabetes and Digestive and Kidney Diseases (NIDDK). The objective of this meeting was to describe recent advances in the field of pancreatic physiopathology and make them available to various medical professionals involved in this area, such as pediatricians, diabetologists, internists, and gastroenterologists [2].

The objective of this narrative review is to gather the most recent data on the topic and make them accessible for specialists, such as pancreatic surgeons and endoscopists. Even though they may not be directly involved in endocrine or exocrine pancreatic disorders, staying updated on pancreatic physiopathology can enhance their clinical practice.

## 2. Materials and Methods

A search on PubMed was carried out using specific terms related to pancreas pathophysiology, including “exocrine insufficiency and pancreatic exocrine insufficiency”, “diabetes and endocrine insufficiency”, “pancreas physiology”, and “pancreatic metabolism diseases”. The decision not to exclude articles based on chronological order allows for a comprehensive overview of the existing literature on this subject.

Considering only reports in the English language with full-text availability and without limitations on article types ensures a diverse set of sources, including original research articles, reviews, and other types of publications, providing a comprehensive understanding of the field.

Additionally, the inclusion of the references reported in the selected articles as other bibliographic sources adds depth to the research, allowing for a more nuanced exploration of the topic.

## 3. Results

### 3.1. The Physiology of the Pancreas

The pancreas is a complex organ that contains both an endocrine and an exocrine component. The exocrine pancreas is responsible for producing, storing, and secreting pancreatic enzymes. The acinar cells produce, store, and secrete digestive enzymes, including amylase, lipase, nucleases, and proteases, such as trypsinogen and chymotrypsin, known in an inactive form as zymogens. The release of these enzymes is stimulated by the hormones cholecystokinin (CCK) and secretin [3,4].

Pancreatic secretion is initiated at the cephalic and gastric phases of digestion, but it is during the intestinal phase that the luminal phase of digestion takes place [5,6]. The latter phase of pancreatic secretion is primarily driven by the release of CCK, a neuroenteroendocrine hormone. CCK is released when the intestinal wall interacts with fatty acids, amino acids, and acidic chyme. In turn, CCK activates the vagal nerve, which lead to zymogen degranulation in acinar cells through cholinergic stimulation [7].

Furthermore, the contact between the acidic chyme and the duodenal mucosa triggers afferent fibers originating from the duodenal wall. These fibers eventually contribute to the development of a vagal-vagal entero-pancreatic reflex. This reflex activates intrapancreatic postganglionic neurons, which play a role in promoting acinar secretion [8]. Additionally, as the acidic chyme enters the duodenum, the pH of the luminal environment decreases. The presence of both low duodenal pH and the stimulus from secretin results in the secretion of sodium, chloride, bicarbonate, and water into the main pancreatic duct, which ultimately flows into the duodenal lumen. This process involves the cystic fibrosis transmembrane conductance channel and aquaporins, and it plays a vital role in facilitating digestion. The secretion of water and bicarbonate serves to elevate the pH of the intestine, thereby stabilizing the activity of pancreatic enzymes [9,10].

In addition to the exocrine pancreas’ role in producing digestive enzymes, the pancreas is also an endocrine organ, playing a crucial role in producing various hormones responsible for blood sugar regulation, among other things. The endocrine pancreas comprises five subtypes of islet cells: alpha, beta, delta, gamma, and epsilon cells. The majority of these cells, accounting for approximately 75%, are beta cells, which are responsible for producing amylin and insulin [4]. Glucagon is produced by alpha cells, somatostatin by delta cells, p-polypeptide by gamma cells, and ghrelin by epsilon cells.

Insulin and glucagon have pivotal effects on maintaining blood glucose homeostasis. Insulin promotes the uptake of blood glucose and its conversion into fatty acids through a process known as lipogenesis. These fatty acids are stored and represent one of the main reserves of energy for the human body [11]. Glucagon, a 29-amino-acid peptide, exhibits a range of biological functions, primarily associated with the regulation of glucose levels. The secretion of glucagon is triggered by multiple metabolic cues, including fluctuations in blood glucose levels, specific amino acids, and potentially free fatty acids. Additionally, stressors like sympathetic nervous system activation can stimulate its release [12].

### 3.2. The Crosstalk between the Endocrine and Exocrine Pancreas

Various hormones produced and released by the endocrine pancreas have an impact on acinar function in the exocrine pancreas. One mode of interaction is cellular trophism. Insulin acts as a trophic factor in the acinar epithelium, stimulating both the synthesis and secretion of amylase while promoting acinar cell growth. The effects of pancreatic polypeptides, somatostatins, and ghrelin on secretion are less well understood, but they may display opposite effects [9,13]. The effect of glucagon on acinar function is bidirectional and largely dependent on the duration of the stimulus. In physiological conditions, when glucagon secretion is pulsatile and influenced by blood glucose levels, its effect on the acini depends on the fasting status. Glucagon has a stimulatory effect during satiety and an inhibitory effect during fasting. Conditions with chronically increased glucagon levels can lead to acinar atrophy, such as in diabetes with insulin deficiency or, less commonly, in glucagonomas [4].

Although it needs further confirmation, a second potential mechanism of interaction between the endocrine and exocrine compartments is gatekeeping. In animal models, it has been suggested that Langerhans islets act as gatekeepers of the exocrine pancreas [14], since anatomically, the blood flow and toxic agents first reach the islets and then the exocrine cells through an islet–acinar portal system [15,16].

Therefore, a speculative theory is that the islets regulate the exposure of the exocrine pancreas to external and metabolic stress factors that play a role in chronic inflammation or cancer development.

Like the liver, pancreatic tissue also contains cytochrome enzymes responsible for metabolism. For example, enzymes involved in type I metabolism, including various CYP isozymes, are distributed throughout the entire pancreas and may participate in detoxifying the bloodstream before it reaches the exocrine pancreas. Certain CYP isoforms, such as CYP2E1, are specifically involved in metabolizing carcinogenic nitrosamines found in cigarette smoke [14,17,18].

Standop J. et al. investigated the prevalence of eight major cytochrome enzymes involved in the detoxification of xenobiotics and alcohol in the parenchyma of a normal pancreas, chronic pancreatitis, and pancreatic cancer. Interestingly, the distribution of cytochromes, although variable in different conditions and diseases, is expressed in immunohistochemistry throughout the pancreas. In chronic pancreatitis tissue, the authors demonstrated a higher presence of cytochromes involved in alcohol detoxification in the islet compartment. The authors concluded that islets, due to their specific microcirculation, might represent a potential site of early activation of CYP enzymes in pancreatic diseases [19].

A third important aspect to consider in the complex paracrine interplay triggered by food ingestion is the simultaneous stimulation of both the exocrine pancreas and the endocrine pancreas.

In fact, the regulation of the endocrine pancreas is also influenced by paracrine stimuli from the gut. Food ingestion triggers the secretion of multiple gut peptides, including the incretin hormones glucose-dependent polypeptide (GIP) and glucagon-like peptide 1 (GLP-1) [20]. GIP and GLP-1 are both released from cells in the small intestine and both stimulate beta cell function and survival, leading to amplified insulin signaling [21,22]. GLP-1 is secreted in a biphasic pattern when consuming foods that are rich in carbohydrates, fats, and amino acids, with the first phase occurring around 10–15 min after a meal and the second phase around 30–60 min later. GLP-1 promotes insulin release in response to elevated glucose levels by stimulating beta cells. Additionally, GLP-1 stimulates the release of glucagon and somatostatin [23]. Constitutive GLP-1 signaling also induces acinar cell proliferation, which in turn leads to increased enzyme levels of amylase and lipase [24].

The activation of incretin receptors leads to increased insulin secretion, beta cell proliferation, and energy storage in adipose tissue. GLP-1 plays a role in regulating appetite and satiety, and its activation has been associated with weight loss [25]. Both GIP and GLP1 are susceptible to degradation by the enzyme dipeptidyl peptidase 4 (DPP-4), which has prompted the development of DPP-4 inhibitors as a treatment for type 2 diabetes mellitus. GLP-1 and GIP connect signals related to nutrient intake and pancreatic insulin secretion [23].

Knop et al. conducted a study to explore whether nutrient uptake, rather than simple contact between food and the intestinal mucosa, is associated with increased GLP-1 secretion. The authors also investigated the hypothesis that pancreatic enzyme replacement therapy (PERT) might affect nutrient uptake, which, in turn, influences the secretion of incretins such as GLP-1 and GIP. In a pilot study involving eight males with chronic pancreatitis treated with PERT and eight matched healthy controls, the authors found a direct relationship between nutrient assimilation and postprandial levels of GLP-1 and GIP. However, they did not observe a significant effect on the beta cell component [26]. These findings warrant further confirmation through larger studies with greater statistical power.

### 3.3. Pancreatic Exocrine Insufficiency

Pancreatic exocrine insufficiency is a consequence of diseases that eventually leads to the loss of pancreatic parenchyma (e.g., chronic pancreatitis or autoimmune pancreatitis), obstruction of the Wirsung duct (e.g., periampullary or pancreatic cancer), inactivation of pancreatic enzymes caused by acid oversecretion (e.g., Zollinger Ellison syndrome), or conditions associated with lower pancreatic secretion (e.g., celiac disease) [27,28,29]. In addition to primary pancreatic causes, it is important to recognize the contribution of extra-pancreatic factors such as surgical interventions that modify the anatomy and coordination of enzymatic secretion with the passage of acidic chyme, neurohormonal changes, and anomalies in the pancreatic duct [30,31,32]. Other extra-pancreatic conditions associated with pancreatic exocrine insufficiency include autoimmune diseases such as Sjögren’s syndrome or celiac disease, infectious diseases like HIV, and inflammatory bowel diseases, particularly ulcerative colitis [29,31,33,34].

Furthermore, cigarette smoking and drugs like somatostatin are linked to a 20% prevalence of pancreatic exocrine insufficiency [35]. Diabetes mellitus (both types 1 and 2) represents one of the most common extra-pancreatic conditions associated with pancreatic exocrine insufficiency [31].

Technically, pancreatic exocrine insufficiency can arise from various conditions that disrupt the sequence of events involved in bringing pancreatic enzymes into contact with food.

Pancreatic exocrine insufficiency is characterized by inadequate pancreatic enzymatic activity in the intestine, resulting in impaired digestion of fatty foods and a reduced uptake of fat-soluble vitamins [30]. It is worth noting that impaired enzyme secretion does not perfectly correlate with symptoms of pancreatic insufficiency, as a certain degree of remnant secretion can compensate for the rest of the organ. Clear symptoms, such as the presence of steatorrhea, might be entirely absent until 90% of the gland’s function is compromised [30]. The acini of the pancreas can sustain normal digestive function even when the greatest part of the functional reserve is depleted [36].

Exocrine pancreatic function can be quantified by fecal elastase-1 measurement, which is further discussed in the Section 3.4.

Pancreatic exocrine insufficiency can lead to weight loss, malnutrition, steatorrhea, fat malabsorption, and an impaired absorption of fat-soluble micronutrients, including vitamins A, D, E, and K [37]. Multiple studies have established a correlation between chronic pancreatitis and exocrine pancreatic insufficiency, as well as an increased risk of developing osteoporosis and osteopenia due to the impaired absorption of vitamin D [37]. Pancreatic exocrine insufficiency is also associated with frailty, cardiovascular disease, infections, cancer, and increased mortality [38].

### 3.4. Testing for Pancreatic Exocrine Insufficiency

The gold standard for assessing pancreatic exocrine function is the fecal elastase test. The test measures the excretion of pancreatic elastase, an enzyme that remains stable throughout the digestive tract until it reaches the intestine [31]. Fecal elastase-1 values above 500 µg/g are considered within the normal range, while values below 200 µg/g indicate severe exocrine pancreatic insufficiency. Values between 200 and 500 µg/g suggest an intermediate level of impairment in exocrine pancreatic secretion [39]. Following pancreatic resections, testing for fecal elastase is limited due to the low sensitivity and specificity of the test. As a result, fecal elastase is not a reliable post-surgical test, and clinicians should always focus their attention on avoiding the undertreatment/underdosage of pancreatic enzyme replacement therapy [40]. It is important to note that fecal elastase testing may have limitations in terms of specificity due to the potential presence of false positives in cases of diarrhea. This is because the enzyme can be diluted in fecal material, leading to potential inaccuracies in the test results [39].

Other methods to measure pancreatic exocrine insufficiency include secretin-enhanced magnetic resonance cholangiopancreatography (S-MRCP) and the C13 mixed triglyceride breath test (C13-MTG) [31,41].

S-MRCP semi-quantitatively assesses the pancreatic exocrine function by measuring the emptying of pancreatic juice into the duodenum [42]. S-MRCP has limited sensitivity and specificity, particularly in the early stages of the disease, as do other semi-quantitative radiological tests. As a result, they have limited clinical utility in managing patients with moderate pancreatic exocrine insufficiency. The C13-MTG breath test measures the excretion of labeled carbon in exhaled air following a meal of labeled triglycerides. A value below 29% is indicative of reduced pancreatic lipase activity, resulting in inadequate triglyceride digestion [39].

### 3.5. Development of Pancreatic Exocrine Insufficiency in Diabetes Mellitus

The presence of pancreatic diseases such as chronic pancreatitis or main pancreatic duct obstruction caused by pancreatic cancer is often associated with the development of type 3c diabetes. This form of diabetes mellitus is also referred to as “pancreatogenic” diabetes and is characterized by simultaneous damage to both the acinar component and the islets due to inflammatory and neoplastic diseases of the pancreas [43]. While a certain relationship between pancreatic diseases and type 3c diabetes has been established, less is known about the effects that diabetes might have on exocrine function. The underlying mechanisms contributing to the development of exocrine insufficiency in patients with type 1 diabetes mellitus are likely multifactorial but not yet fully understood. These may include the lack of trophic action of insulin, glucagon, and somatostatin, autoimmune damage to the pancreas, and the presence of autonomic diabetic neuropathy, which can impair the entero-pancreatic reflex.

Additionally, generalized pancreatic inflammation characterized by chronic inflammatory infiltrate and microvascular damage leading to hypoxic damage can also contribute to chronic insult to the exocrine component.

Rodriguez-Calvo et al. collected cadaveric pancreas samples from type 1 diabetes patients, type 2 diabetes patients, diabetes-free subjects with islet autoantibodies, and healthy controls. The authors investigated the presence of CD8+, CD4+, and CD11c+ cell infiltration, showing a consistent difference in the CD8+ infiltrate between healthy individuals and diabetic patients. Moreover, the diabetic-free subjects’ islet autoantibodies displayed a considerable infiltrate of CD4+ and CD11c+ cells, indicating a potential role of such populations in the early stage of the disease [44].

Early onset, significant insulin shortage, prolonged illness duration, and high rates of neurological and vascular consequences are typical characteristics of type 1 diabetes. These elements all have a high likelihood of causing the exocrine pancreas to become chronically damaged, which will ultimately lead to a reduction in exocrine function [45].

The pathophysiological mechanisms leading to pancreatic exocrine insufficiency in type 2 diabetes mellitus are similar to those described for type 1 diabetes. Autonomic neuropathy and microvascular damage contribute to pancreatic atrophy and fibrosis in individuals with type 2 diabetes, particularly in the absence of autoimmune damage and severe insulin deficiency [46]. Chronic microvascular and neuronal complications of diabetes damage the exocrine pancreas, like the parenchymal changes observed in other susceptible organs. Consequently, pancreatic exocrine insufficiency is more common in severe or poorly controlled diabetes, especially in the presence of autonomic neuropathy or existing microvascular damage. These pathogenic insults may promote the development of fibrosis and atrophy, which are exacerbated by the interruption of the islet–acinar–ductal axis. The islet–acinar axis refers to the interactions between the endocrine pancreas and the exocrine pancreas, exemplified by the influence of exocrine secretions on beta cell function and endocrine hormones’ effect on pancreatic secretions. The pancreas has a unique vascular structure in which the islets receive a greater proportion of arterial blood compared with the exocrine pancreas. The concentration of insulin in the pancreatic peri-islet tissue is much higher than in the systemic circulation. For example, despite rapid dilution, the concentration of insulin in the portal stream is twice that in the peripheral blood. Insulin exhibits a trophic action on the acinar epithelium and can modulate pancreatic enzyme secretion [47].

The trophic effect of insulin on the acinar component has already been suggested in early studies by Mössner J. et al. The authors proved the presence of insulin receptors by testing the presence of 125l-labeled hormones (iodinated IGF-I, IGF-II, and insulin) on cultured cells (AR42J) derived from transplantable tumors of the acinar compartment in rat models. The authors showed that an increasing concentration of insulin from 1 nM to 100 nM resulted in a linear increase in both the synthesis of DNA and cell growth. Specifically, cell growth was proven to be increased by 46.1 ± 10.9% for an insulin concentration equivalent to 100 nM [48]. Moreover, the same group has shown a decrease in the amount of insulin receptors on acinar cells in rat models of diabetics after the experimental administration of streptozotocin, with a 50% decrease in 125I-labeled insulin uptake from the acinar component, despite a conserved receptor affinity as measured by IC50 [49].

Besides the trophic effect of insulin on the acinar component, it is possible that another potential mechanism that links endocrine and exocrine insufficiency is based on a direct toxic effect of hyperglycemia. In fact, Nomiyama Y. et al. have investigated in rat models the role of increasing glucose concentration on the activation and proliferation of pancreatic stellate cells, the activation of α-smooth muscle actin, and the production of collagen. The authors proved that high glucose concentrations caused increased activation of the protein kinase C and mitogen-activated protein kinase-driven signaling, which eventually led to increased pancreatic fibrosis [50].

### 3.6. The Prevalence of Pancreatic Exocrine Insufficiency in Type 1 Diabetes

Type 1 diabetes is associated with exocrine pancreatic insufficiency and a pronounced degree of parenchymal atrophy. In patients with type 1 diabetes, the pancreatic volume as well as the pancreatic weight are reduced by 19–50% and 55%, respectively, when compared to patients with type 2 diabetes and healthy individuals [3,4].

Children with diabetes mellitus also have a reduction in pancreatic volume ranging from 16 to 61% [51,52]. Interestingly, the large majority of pancreatic weight loss appears to happen during the first phase of diabetes mellitus [53]. Microscopic analysis reveals the presence of fibrosis, fatty degeneration, defined as a proportional increase in the fat content in a gland, and inflammatory infiltrates in the pancreas of patients with both type 1 and type 2 diabetes [54]. These features are commonly observed in patients with chronic pancreatitis and exocrine insufficiency as well. In contrast to chronic pancreatitis, fibrosis in a diabetic pancreas is typically described as intra-lobular/intra-acinar with signs of acinar atrophy, and the ductal component remains intact. Approximately 30% of patients with type 1 and type 2 diabetes have fatty infiltration of the pancreas, 47% have lymphocytic infiltration, and microvascular changes are present in almost all patients upon autopsy, manifesting as a wall thickening of small arterioles [3]. In children with type 1 diabetes mellitus, the prevalence of severe pancreatic exocrine insufficiency is approximately 10%, while moderate pancreatic exocrine insufficiency is present in 45% of cases. A study conducted by Larger et al. examined the presence of pancreatic exocrine insufficiency in a cohort of patients with diabetes mellitus. Among the 95 patients with type 1 diabetes, the prevalence of severe pancreatic exocrine insufficiency, defined as a fecal elastase-1 concentration below 200 μg/g, was observed in 34% of cases [45,55,56].

Overall, it is anticipated that pancreatic exocrine insufficiency is severe in approximately 10–30% of adults with type 1 diabetes, while moderate pancreatic exocrine insufficiency is present in around 22–56% of those patients [56].

Pancreatic exocrine insufficiency in type 1 diabetes can manifest as symptoms such as diarrhea, abdominal bloating, cramping pains, and weight loss [21]. Symptoms of pancreatic exocrine insufficiency in type 1 diabetes often overlap with those of other diabetes complications, including autonomic neuropathy, celiac disease, and bacterial infections. In particular, celiac disease in patients with type 1 diabetes displays a prevalence as high as 25% [57].

The clinical significance of pancreatic exocrine insufficiency in type 1 diabetes patients is not well understood and is an underdiagnosed condition. In a prospective observational study, a group of 115 patients with type 1 and type 2 diabetes mellitus and exocrine pancreatic insufficiency were randomly assigned to receive pancreatic exocrine replacement therapy (PERT) or a placebo. Among these patients, 42% reported gastrointestinal symptoms [58]. The study concluded that pancreatic exocrine replacement therapy (PERT) did not have a significant effect on symptoms in patients with pancreatic exocrine insufficiency and diabetes mellitus. It is important to note that the follow-up interval in the study was only three months. Additionally, no statistically significant differences were observed in HbA1c levels, fasting glucose levels, or glucose levels two hours after a meal between the PERT group and the placebo group. The authors in fact suggested a possible reduction in mild and moderate hypoglycemia [58].

### 3.7. The Prevalence of Pancreatic Exocrine Insufficiency in Type 2 Diabetes

Data on the prevalence of pancreatic exocrine insufficiency in adult patients with type 2 diabetes are abundant, although they exhibit heterogeneity. In a study by Larger E et al., a cohort of 472 patients with type 2 diabetes reported a prevalence of severe pancreatic exocrine insufficiency of 20% [56].

Lidkvist et al. conducted a study to examine fecal elastase levels and nutritional status in 315 patients with type 2 diabetes mellitus. The study reported a prevalence of severe exocrine insufficiency of 5.2%, while the prevalence of mild exocrine insufficiency was found to be 4.9% [59].

In a large prospective study involving 1231 diabetic patients, the prevalence of pancreatic exocrine insufficiency was reported to be 35% among a subgroup of 697 individuals with type 2 diabetes. However, the study did not find any significant associations between disease duration and insulin therapy in the type 2 diabetes subgroup, contrary to the observations made in the overall population of 1231 participants. This suggests that factors other than disease duration and insulin therapy may contribute to the development of pancreatic exocrine insufficiency in individuals with type 2 diabetes [46].

It is important to consider that the aforementioned study did not exclude individuals with a history of pancreatic disease, which could have influenced the observed prevalence of pancreatic exocrine insufficiency in the type 2 diabetes subgroup. The presence of pre-existing pancreatic conditions could potentially contribute to a higher prevalence of pancreatic exocrine insufficiency in individuals with type 2 diabetes. Subsequent, smaller series of studies have confirmed that the prevalence of severe pancreatic exocrine insufficiency (defined as fecal elastase of <100) ranges between 12% and 21% in patients with type 2 diabetes. For example, in a cross-sectional study by Hardt et al. involving 697 patients, a prevalence of fecal elastase of <100 was reported in 19.9% of patients [45]. Similarly, in another cross-sectional study conducted by Ewald et al. involving 546 patients, a prevalence of fecal elastase of <100 was reported in 21% of the patients [54]. Both Rathmann et al. and Larger et al. report a 12% prevalence of severe pancreatic exocrine insufficiency (elastase < 100) in type 2 diabetes patients [56,60].

## 4. Limitations

While our study aims to provide a comprehensive overview with a high level of scientific accuracy, it inherently possesses some limitations commonly associated with non-systematic literature reviews. Firstly, the presented study offers an extensive examination of the physiopathology of the pancreas, lacking a hypothesis specifically focused on addressing a singular aspect of this research topic. Secondly, the literature search was exclusively conducted using PubMed. Thirdly, the results of various articles have been presented without additional analysis. Fourth, our review did not thoroughly address all the molecular details involved in the exocrine–endocrine relationship. Lastly, it is acknowledged that this type of study may be somewhat influenced by the authors’ perspectives and intentions. 

## 5. Conclusions

The endocrine pancreas and exocrine pancreas have a reciprocal interaction, and diseases affecting one can impact the other. Although the prevalence of pancreatic exocrine insufficiency in type 1 and type 2 diabetes has been examined in cross-sectional studies, the incidence of the disease during the course of diabetes is not well described. Approximately half of patients with type 1 diabetes and one-third of patients with type 2 diabetes exhibit exocrine insufficiency, and this insufficiency is often associated with a longer duration of disease and poorer glycemic control. Further research is needed to determine the precise prevalence of pancreatic exocrine insufficiency in individuals with diabetes and to evaluate whether pancreatic enzyme replacement therapy is able to enhance glycemic control and reduce the risk of potentially life-threatening hypoglycemic episodes.

## Data Availability

Not applicable.

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
