# Peer review of "Interactions between the Exocrine and the Endocrine Pancreas"

_jcm, 2024, doi:10.3390/jcm13041179_

Round 1

Reviewer 1 Report

Comments and Suggestions for Authors

The manuscript is a narrative review, extremely well-written, based on thorough research revealing the paracrine interaction between the pancreatic islets and acini. Moreover it makes light in a matter poorly understood in digestive pathology, the relationship between diabetes mellitus and exocrine insufficiency, we have encountered this even in pancreatic cancer, but an objective correlation was not found until now, therefore I recommend this paper for Minor revisions.

Comments on the Quality of English Language

Minor spelling and grammar inconsistencies need chech-up.

Author Response

We would like to thank the Reviewer for their suggestions and comments, which have allowed us to critically revise our work. We have attempted to improve our manuscript addressing all of their comments, and we believe that, in doing so, the work has significantly improved.

Thank you once again for the time dedicated to reviewing our work, and we hope that the changes made are satisfactory to the Reviewer.

All references have been modified and renumbered.

Reviewer 2 Report

Comments and Suggestions for Authors

Dear authors

The paper is superficial. It discusses macro-physiology but omits molecular mechanisms involved in the exo-endocrine relationship. It has the level of a textbook for undergraduates. Otherwise, it is easy to read.

Figure 1 is not good and gives no information. I would suggest to eliminate it.

You say:

A second potential mechanism of interaction between endocrine and exocrine compartment is the gatekeeping. In fact, the islets in the endocrine pancreas can regulate the exposure of the exocrine pancreas to external and metabolic stress factors that play a role in chronic inflammation or cancer development. It has been suggested that Langerhans islets act as gatekeepers of the exocrine pancreas, [12] since anatomically the blood flow first reaches the islets, and then the exocrine cells. [13].

This idea is not sufficiently backed by the references. Ref 12 is a review and ref 13 does not say what you are suggesting. You should clearly say that this is your speculation, and not an established fact.

You say

Like the liver, pancreatic tissue also contains cytochrome enzymes responsible for metabolism. For example, enzymes involved in type I metabolism, including various CYP 98 isozymes, are distributed throughout the entire pancreas and may participate in detoxifying the bloodstream before it reaches the exocrine pancreas.

Please, clarify where these enzymes are. How can they detoxify the bloodstream before it reaches the exocrine pancreas. To the best of my knowledge these enzymes are in the acini.

You say

In fact, the regulation of the endocrine pancreas is also influenced by paracrine stimuli  from the gut. Food ingestion triggers the secretion of multiple gut peptides including the incretin hormones glucose-dependent polypeptide (GIP) and glucagon-like peptide 1 (GLP-1). GIP and GLP-1 are both released from cells in the small intestine and both stimulate beta cell function and survival leading to amplified insulin signaling.16 GLP-1 is secreted in a biphasic pattern when consuming foods that are rich in carbohydrates, fats and amino acids, with the first phase occurring around 10-15 minutes after the meal and the second phase around 30-60 minutes later. GLP-1 promote insulin release in response to elevated glucose levels by stimulating beta cells.

Too many concepts without a reference. Please add.

You say

The activation of incretin receptors leads to increased insulin secretion, beta-cell proliferation, and energy storage in adipose tissue. GLP-1 plays a role in regulating appetite and satiety and its activation has been associated with weight loss.

This phrase needs a reference

Figure 2 is a very poor quality figure. I suggest to eliminate it. In case of being replaced it should be a figure that gives more information than what now has. Such a poor information does not need a figure.

For the Pancreatic exocrine insufficiency I suggest you complete it with concepts from

DomínguezMuñoz, J. E. (2011). Pancreatic exocrine insufficiency: diagnosis and treatment. Journal of gastroenterology and hepatology26, 12-16.

Author Response

We would like to thank the Reviewer for their suggestions and comments, which have allowed us to critically revise our work. We have attempted to address all of their comments, and we believe that, in doing so, the work has significantly improved.

Thank you once again for the time dedicated to reviewing our work, and we hope that the changes made are satisfactory to the reviewer.

All references have been modified and renumbered.

Comment: The paper is superficial. It discusses macro-physiology but omits molecular mechanisms involved in the exo-endocrine relationship. It has the level of a textbook for undergraduates. Otherwise, it is easy to read.

Figure 1 is not good and gives no information. I would suggest to eliminate it.

Answer: we thank the Reviewer for their comment. As suggested we have removed Figure 1 from the manuscript.

Comment: You say:

A second potential mechanism of interaction between endocrine and exocrine compartment is the gatekeeping. In fact, the islets in the endocrine pancreas can regulate the exposure of the exocrine pancreas to external and metabolic stress factors that play a role in chronic inflammation or cancer development. It has been suggested that Langerhans islets act as gatekeepers of the exocrine pancreas, [12] since anatomically the blood flow first reaches the islets, and then the exocrine cells. [13].

This idea is not sufficiently backed by the references. Ref 12 is a review and ref 13 does not say what you are suggesting. You should clearly say that this is your speculation, and not an established fact.

Answer: we thank the Reviewer for their valuable comment. We have clarified that it is a speculative theory in need for further confirmations, and we have added a further reference based on animal models showing how a toxic damage might progress from the islet-peri islet area and eventually to the acinar component.

Comment: You say

Like the liver, pancreatic tissue also contains cytochrome enzymes responsible for metabolism. For example, enzymes involved in type I metabolism, including various CYP 98 isozymes, are distributed throughout the entire pancreas and may participate in detoxifying the bloodstream before it reaches the exocrine pancreas.

Please, clarify where these enzymes are. How can they detoxify the bloodstream before it reaches the exocrine pancreas. To the best of my knowledge these enzymes are in the acini.

Answer: Thanks for your interesting question. Standop J et al. investigated the prevalence of eight major cytochrome enzymes involved in the detoxification of xenobiotics and alcohol in the parenchyma of the normal pancreas, chronic pancreatitis, and pancreatic cancer. Interestingly, the distribution of cytochromes, although variable in different conditions and diseases, is expressed in immunohistochemistry throughout the pancreas. In chronic pancreatitis tissue, the authors demonstrated a higher presence of cytochromes involved in alcohol detoxification in the islet compartment. We have incorporated this concept and added a reference

Comment: You say

In fact, the regulation of the endocrine pancreas is also influenced by paracrine stimuli  from the gut. Food ingestion triggers the secretion of multiple gut peptides including the incretin hormones glucose-dependent polypeptide (GIP) and glucagon-like peptide 1 (GLP-1). GIP and GLP-1 are both released from cells in the small intestine and both stimulate beta cell function and survival leading to amplified insulin signaling.16 GLP-1 is secreted in a biphasic pattern when consuming foods that are rich in carbohydrates, fats and amino acids, with the first phase occurring around 10-15 minutes after the meal and the second phase around 30-60 minutes later. GLP-1 promote insulin release in response to elevated glucose levels by stimulating beta cells.

Too many concepts without a reference. Please add.

Answer: We thank the reviewer for their comment. We have added the references as rightly suggested by the reviewer, whom we appreciate.

Comment: You say

The activation of incretin receptors leads to increased insulin secretion, beta-cell proliferation, and energy storage in adipose tissue. GLP-1 plays a role in regulating appetite and satiety and its activation has been associated with weight loss.

This phrase needs a reference

Answer: We thank the reviewer for their comment. We have added the references as rightly suggested by the reviewer, whom we sincerely appreciate.

Comment: Figure 2 is a very poor quality figure. I suggest to eliminate it. In case of being replaced it should be a figure that gives more information than what now has. Such a poor information does not need a figure.

Answer: We thank the reviewer for their comment. We have removed Figure 2 as  suggested by the reviewer.

Comment: For the Pancreatic exocrine insufficiency I suggest you complete it with concepts from

Domínguez‐Muñoz, J. E. (2011). Pancreatic exocrine insufficiency: diagnosis and treatment. Journal of gastroenterology and hepatology26, 12-16.

Answer: We thank the reviewer for their suggestion. We have added the reference above as rightly suggested by the reviewer. We really appreciate their suggestion that for sure improve our paper.

Round 2

Reviewer 2 Report

Comments and Suggestions for Authors

Dear authors

I recommend you remove the paragraph on limitations. The fact that you are surgeons is a poor excuse for writing a paper that does not go into the molecular details involved in the exocrine-endocrine relationship.

You omit intracellular pathways involved in this relationship. So that the level is still that of an undergraduate textbook.

Author Response

We thank the Reviewer once again for their commitment and the time they have dedicated to our work. Their valuable comments have allowed us to further improve our manuscript.

The added parts are highlighted in green, while the parts to be eliminated are crossed out. The entire bibliography has been renumbered based on the additions made.

We hope that the new version will be appreciated by the Reviewer, whom we thank for their very important contribution.

Comment: I recommend you remove the paragraph on limitations. The fact that you are surgeons is a poor excuse for writing a paper that does not go into the molecular details involved in the exocrine-endocrine relationship.

Answer: We thank the Reviewer for their suggestions and comments. As rightly pointed out, we have modified the "study limitations" paragraph by removing the part suggested by the Reviewer.

Additionally, following the invaluable advice of the Reviewer, we have added two paragraphs (highlighted in green) to include in the review some aspects of the molecular details involved in the exocrine-endocrine relationship. Furthermore, in the limitations section, we have added that a limitation of the review is the fact that we did not delve deeply into all the molecular mechanisms involved in the exocrine-endocrine relationship.